# Enhancing Clinical Data Analysis by Explaining Interaction Effects between Covariates in Deep Neural Network Models

**DOI:** 10.3390/jpm13020217

**Published:** 2023-01-26

**Authors:** Yijun Shao, Ali Ahmed, Edward Y. Zamrini, Yan Cheng, Joseph L. Goulet, Qing Zeng-Treitler

**Affiliations:** 1Department of Clinical Research and Leadership, School of Medicine and Health Sciences, George Washington University, Washington, DC 20037, USA; 2Washington DC VA Medical Center, Washington, DC 20422, USA; 3Department of Medicine, School of Medicine, Georgetown University, Washington, DC 20057, USA; 4Department of Neurology, School of Medicine, University of Utah, Salt Lake City, UT 84108, USA; 5Irvine Clinical Research, Irvine, CA 92614, USA; 6Cognitive Neurology Consulting, Newport Beach, CA 92614, USA; 7VA Connecticut Healthcare System, New Haven, CT 06516, USA; 8Department of Emergency Medicine, Yale School of Medicine, Yale University, New Haven, CT 06516, USA

**Keywords:** deep learning, risk analysis, Alzheimer’s disease and related dementia

## Abstract

Deep neural network (DNN) is a powerful technology that is being utilized by a growing number and range of research projects, including disease risk prediction models. One of the key strengths of DNN is its ability to model non-linear relationships, which include covariate interactions. We developed a novel method called interaction scores for measuring the covariate interactions captured by DNN models. As the method is model-agnostic, it can also be applied to other types of machine learning models. It is designed to be a generalization of the coefficient of the interaction term in a logistic regression; hence, its values are easily interpretable. The interaction score can be calculated at both an individual level and population level. The individual-level score provides an individualized explanation for covariate interactions. We applied this method to two simulated datasets and a real-world clinical dataset on Alzheimer’s disease and related dementia (ADRD). We also applied two existing interaction measurement methods to those datasets for comparison. The results on the simulated datasets showed that the interaction score method can explain the underlying interaction effects, there are strong correlations between the population-level interaction scores and the ground truth values, and the individual-level interaction scores vary when the interaction was designed to be non-uniform. Another validation of our new method is that the interactions discovered from the ADRD data included both known and novel relationships.

## 1. Introduction

As a key artificial intelligence method, deep neural networks (DNNs) demonstrated break-through performances in a variety of tasks such as computer vision, speech recognition and board game playing [1,2,3]. Inspired by such successes, researchers applied DNNs to biomedical research [4,5,6,7]. The specific use cases range from image recognition, clinical risk prediction and disease diagnosis systems to natural language processing and identification of genes associated with disease.

On the other hand, DNN models are known to be difficult to explain compared to traditional statistical models such as linear regression, which limits their utilization and adoption in many areas such as in clinical practice. Building a more accurate prediction or classification system is not always the goal; in clinical research, investigators are often equally, if not more, interested in the relationship between the predictors (e.g., demographic or clinical characteristics) and the outcome. It is thus important to develop explanation methods for DNN models.

Approaches for explaining DNN models include Attention Mechanism [8], Local Interpretable Model-agnostic Explanations (LIME) [9], SHapley Additive exPlanations (SHAP) [10,11] and Layer-wise Relevance Propagation (LRP) [12]. In a similar spirit as LIME, we have developed a method called "impact scores" to explain the DNN models [13]. Impact scores quantify the impact of individual variables on the outcome. They generalize the coefficients of the predictor variables in linear logistic regression models to DNN models.

All the aforementioned methods are designed to explain the effects of the individual variables on the outcome, which together constitute the major linear part of the total effect. In reality, the predictor–outcome relationship is almost always nonlinear, and interactions need to be considered for explaining the remaining effects. According to Berrington de González and Cox [14], "From the statistical perspective, interaction is said to occur if the separate effects of the factors do not combine additively". Covariate interactions are a topic of interest in healthcare studies. For example, smoking and exposure to asbestos have an interaction effect on the risk of lung cancer [15]. Covariate interactions are clearly defined in the context of regression [16], but they are less well studied in the context of DNN models. Understanding the interactions captured by DNN could provide new insights into biological or clinical relationships. 

Different methods have been developed to understand or quantify the interaction effects in DNN models. These methods include Intrator’s graphical tool [17], Neural Interaction Transparency (NIT) [18], Bayesian Group Expected Hessian (GEH) [19], KL-diff^2^ [20], SHAP interaction value [21], Global Surrogate Model (GSM) interaction coefficient [22], and Partial Dependence Variation (PDV) interaction value [23]. However, each has certain limitations, such as the results not being straightforward to interpret (e.g., KL-diff^2^), or the method relying on a particular type of DNN model (e.g., NIT, GEH), or the computation being expensive (e.g., SHAP interaction). In Table 1, we summarize the comparison of these methods in various aspects including what the approach is, whether it is model-agnostic, and how much the computational cost is.

In this study, we develop "interaction scores" to quantify the covariate interactions in DNN models predicting binary outcomes, which can be viewed as the generalizations of the interaction coefficients from regression models to DNN models. The advantage of the method is that the interpretations of the interaction scores are similar or even the same as those of the regression interaction coefficients that the statisticians and clinicians are already familiar with. In addition, the method is model-agnostic, meaning that it works without knowing the inner structure of the model. Therefore, the method can be applied to any machine learning model with the same input and output format as the DNN models.

This method is applied to two DNN models trained on three datasets, respectively; two simulated and one real-world clinical dataset. We use simulated data because the exact relationship between the covariates and outcome is known, which allows us to evaluate the computed interaction scores against the true interaction effects. The clinical dataset is derived from a project on the racial disparity in incident Alzheimer’s disease and related dementias (ADRD) [24]. We are particularly interested in the interaction between race and other risk factors. This dataset helps to demonstrate the value of interaction scores in real-world settings.

## 2. Materials and Methods

Consider a DNN with an input layer with n nodes, an output layer with one node, and several hidden layers with various numbers of nodes each. The activation functions for the hidden layers are commonly chosen as the rectified linear unit (ReLU) function ρ(x)=max(0,x) but can also be other functions such as the tanh. The activation function for the output layer, however, is always the sigmoid function $\sigma \left(x\right)=e^{x}/\left(1+e^{x}\right)$, so that the output is a single value between 0 and 1. This type of DNN is typically used for predicting binary outcomes such as mortality. Let $p=F\left(x_{1},\ldots ,\;x_{n}\right)$ denote the final DNN model, where $x_{1},\ldots ,\;x_{n}$ are the n variables corresponding to the n nodes of the input layer and p (0<p<1) is the output of the model representing the risk of the adverse outcome. Specifically, if z is the outcome variable taking values 0 and 1 with 1 representing the adverse outcome, then p=P(z=1).

For the purpose of defining impact scores and interaction scores, we first choose a fixed value $x_{j}^{r}$ for each variable $x_{j}$, and call it the reference value of $x_{j}$. The reference values will serve as baselines to which other values (or situations) can be compared. For example, for binary variables with values 0 and 1, representing absence and presence of a diagnosis, respectively, usually the value 0 (i.e., absence of the diagnosis) is chosen as the reference value. The general guideline is to choose the most "common" (e.g., mean, median, mode) values of the variable as the reference value, but it is not a strict rule and one is free to choose other values as the reference value relevant within the context. 

Next, we define $f\left(x_{1},\ldots ,\;x_{n}\right)=\mathrm{logit}\left(F\left(x_{1},\ldots ,\;x_{n}\right)\right)$, where $\mathrm{logit}\left(x\right)=\log\frac{x}{1-x}\;$ is the inverse function of the sigmoid function σ. The output value of f ranges from −∞ to ∞. Taking an individual subject whose value for $x_{j}$ is denoted as $x_{j}^{c}$ for j=1,…,n, we define the impact score of the variable $x_{j}$ on this individual subject as

$$\mathrm{Impact}\;\mathrm{score}=\frac{f\left(\cdots ,x_{j}^{c},\cdots \right)-f\left(\cdots ,x_{j}^{r},\cdots \right)}{x_{j}^{c}-x_{j}^{r}}$$ where "⋯" represents the values of this individual for all variables other than $x_{j}$. This is called the individual-level impact score. Note that it is only defined for the subjects with $x_{j}^{c}\neq x_{j}^{r}$. The populational-level impact score of $x_{j}$ is defined as the mean of all the individual-level impact scores of $x_{j}$.

The expression of the impact score is a ratio—the numerate is the change of the logit(p), and the denominator is the change in x. Therefore, the impact score is the (average) rate of change of logit(p) with respect to the change in x. Impact scores can be applied to any machine learning model which predicts a binary outcome with a probability value p. Thus, they can be applied to DNN models as well.

As an example, we calculate the impact score for a linear logistic regression model: $$p=F\left(x_{1},\ldots ,\;x_{n}\right)\equiv \sigma \left(\beta_{0}+\beta_{1}x_{1}+\cdots +\beta_{n}x_{n}\right)$$ First, we have $f\left(x_{1},\ldots ,\;x_{n}\right)=\beta_{0}+\beta_{1}x_{1}+\cdots +\beta_{n}x_{n}$. Then, the individual-level impact score of $x_{j}$ is $$\frac{(\beta_{0}+\beta_{1}x_{1}^{c}+\cdots +\beta_{n}x_{n}^{c})-\left(\beta_{0}+\beta_{1}x_{1}^{c}+\cdots +\beta_{j}x_{j}^{r}+\cdots +\beta_{n}x_{n}^{c}\right)}{x_{j}^{c}-x_{j}^{r}}\;=\frac{\beta_{j}x_{j}^{c}-\beta_{i}x_{j}^{r}}{x_{j}^{c}-x_{j}^{r}}=\frac{\beta_{j}(x_{j}^{c}-x_{j}^{r})}{x_{j}^{c}-x_{j}^{r}}=\beta_{j}$$ and the population-level impact score of $x_{j}$ is also $\beta_{j}$. Thus, we have recovered the logistic regression coefficient of the variable $x_{j}$. This example shows that the (population-level) impact score is a generalization of the coefficients of the linear logistic regression models to the DNN models. It also shows why we need to apply the "logit" function to the probability p before we calculate the impact score; without it, we would not recover the regression coefficient on the linear logistic model.

Note that the impact score of a variable does not represent the importance of the variable and is not intended for variable selection. This is similar to the situation of a logistic regression; the impact scores are analogous to the odds ratios, while the variable importance is analogous to the statistical significances characterized by p-values, and in this situation variable selection is not based on odds ratios but on statistical significances.

For two variables $x_{j}$ and $x_{k}$, we define the individual-level interaction score between them on an individual subject as:$$\mathrm{Interaction}\;\mathrm{score}=\frac{f\left(\cdots ,x_{j}^{c},\cdots ,x_{k}^{c},\cdots \right)-f\left(\cdots ,x_{j}^{r},\cdots ,x_{k}^{c},\cdots \right)-f\left(\cdots ,x_{j}^{c},\cdots ,x_{k}^{r},\cdots \right)+f\left(\cdots ,x_{j}^{r},\cdots ,x_{k}^{r},\cdots \right)}{\left(x_{j}^{c}-x_{j}^{r}\right)\left(x_{k}^{c}-x_{k}^{r}\right)}$$ for each individual subject whose values satisfy $x_{j}^{c}\neq x_{j}^{r}$ and $x_{k}^{c}\neq x_{k}^{r}$. The population-level interaction score is similarly defined as the mean of all individual-level interaction scores.

Again, as an example, we calculate the interaction score for a logistic regression model of two variables with an interaction term:$$p=\sigma \left(a+bx_{1}+cx_{2}+dx_{1}x_{2}\right)$$ where $dx_{1}x_{2}$ is the interaction term and d is the interaction coefficient. Applying the interaction score formula, we find the individual-level interaction score between $x_{1}$ and $x_{2}$ to be $$\frac{\left(a+bx_{1}^{c}+cx_{2}^{c}+dx_{1}^{c}x_{2}^{c}\right)-\left(a+bx_{1}^{r}+cx_{2}^{c}+dx_{1}^{r}x_{2}^{c}\right)-\left(a+bx_{1}^{c}+cx_{2}^{r}+dx_{1}^{c}x_{2}^{r}\right)+\left(a+bx_{1}^{r}+cx_{2}^{r}+dx_{1}^{r}x_{2}^{r}\right)}{\left(x_{1}^{c}-x_{1}^{r}\right)\left(x_{2}^{c}-x_{2}^{r}\right)}\;=\frac{dx_{1}^{c}x_{2}^{c}-dx_{1}^{r}x_{2}^{c}-dx_{1}^{c}x_{2}^{r}+dx_{1}^{r}x_{2}^{r}}{\left(x_{1}^{c}-x_{1}^{r}\right)\left(x_{2}^{c}-x_{2}^{r}\right)}=\frac{d\left(x_{1}^{c}-x_{1}^{r}\right)\left(x_{2}^{c}-x_{2}^{r}\right)}{\left(x_{1}^{c}-x_{1}^{r}\right)\left(x_{2}^{c}-x_{2}^{r}\right)}=d$$
Hence, the population-level interaction score is also d. This example shows that the (population-level) interaction score is a generalization of the interaction coefficient in the logistic regression model to the DNN models; hence, the name of "interaction score" is justified.

We apply the interaction scores to two simulated and one real-world clinical dataset. The advantage of simulated data over real-world data is that the true underlying relationship between the predictors and outcome is known, so we can validate the interaction scores on the simulated data. To illustrate the difference between interaction scores and other explaining methods, we also apply two existing methods—the PDV interaction value method and the GSM interaction coefficient method—to all three datasets.

Simulation Data #1. In this simulation, 100 variables, $x_{1},\;x_{2},\;\ldots ,\;x_{100}$, are used as predictors and a single variable z as the outcome. Among the 100 variables, the first 50 are binary variables with values 0/1 and the remaining 50 are continuous variables taking values between 0 and 1. A nonlinear relationship between the log odds of the outcome z=1 and 100 variables $x_{1},\;x_{2},\;\ldots ,\;x_{100}$ is as follows:$$\mathrm{logit}\left(P\left(z=1\right)\right)=\beta_{0}+\overset{100}{\underset{i=1}{\sum }}\beta_{i}x_{i}+\overset{20}{\underset{m=1}{\sum }}\gamma_{m}x_{i_{m}}^{2}+\overset{20}{\underset{m=1}{\sum }}\theta_{m}x_{j_{m}}x_{k_{m}}$$ where ${\left\{i_{m}\right\}}_{m=1}^{20}$, ${\left\{j_{m}\right\}}_{m=1}^{20}$ and ${\left\{k_{m}\right\}}_{m=1}^{20}$ are three randomly sampled subsets of the index set [1, 2, …, 100] with ${\left\{j_{m}\right\}}_{m=1}^{20}\cap^{\;}{\left\{k_{m}\right\}}_{m=1}^{20}=\emptyset$.

There are 20 square terms ($\gamma_{m}x_{i_{m}}^{2}$) and 20 cross-product terms ($\theta_{m}x_{j_{m}}x_{k_{m}}$) in addition to 100 linear terms ($\beta_{i}x_{i}$) and one constant term ($\beta_{0}$) in this relationship. The coefficients $\beta_{i}$’s and $\gamma_{m}$’s are randomly sampled from a uniform distribution with range [−1,1] and $\theta_{m}$’s are randomly generated from a uniform distribution with range [−10,10]. The cross-product terms $\theta_{m}x_{j_{m}}x_{k_{m}}$ are the interaction terms, and $\theta_{m}$ are the interaction coefficients. To generate one sample of data, random values are sampled for the 100 variables $x_{i}$, then a value p=P(z=1) is calculated based on the above relationship, and finally the outcome variable z is randomly sampled as z ~ Bernoulli(p). Repeating this process, we generate a set of 50,000 samples, which are the first simulation data.

This dataset is split into a training (60%), a validation (20%) and a testing set (20%). Then, a DNN model with 10 hidden layers is trained on the training set, with validation AUC observed after each epoch. Early stopping strategy is adopted to avoid over-fitting. Then, the final model is applied to the testing set to calculate the testing AUC.

The (population-level) interaction scores, PDV interaction values and GSM interaction coefficients are calculated based on the trained DNN model and are compared to the interaction coefficients in the nonlinear relationship underlying the simulation data. The comparison uses several metrics, including Pearson’s correlation, Spearman’s rank correlation, and sign agreement. The correlations measure the agreement in relative values and in relative ranks, respectively. The sign agreement is the proportion of the interaction scores with the same sign as the corresponding interaction coefficients. The calculation of the correlations and sign agreements are limited to the 20 pairs of variables included in the nonlinear relationship.

Simulation Data #2. In this simulation, we use a more nonlinear relationship than in simulation #1. This simulation uses four variables, $x_{1},\;x_{2},\;x_{3},\;x_{4}$, as predictors and a variable z for binary outcomes represented by values 0 and 1, respectively. All of the four variables are continuous variables, taking values between –101 and 1. A nonlinear relationship between the log odds of the outcome z=1 and the four variables are given by
$$\begin{array}{cc} \mathrm{logit}\left(P\left(z=1\right)\right) & =-2+\sin\left(\pi x_{1}\right)+\cos\left(\pi x_{2}\right)+e^{x_{3}}+\log\left(x_{4}+1.5\right)\\ & +5x_{1}x_{2}\\ & +5e^{x_{2}}\sin\left(0.5\pi x_{3}\right)-10x_{3}x_{4} \end{array}$$ To generate one sample of data, random values are sampled for the four variables first, then a value p=P(z=1) is calculated based on the above relationship, and finally the outcome variable z is randomly sampled as z ~ Bernoulli(p). Repeating this process, we generate a set of 50,000 samples, which are the second simulation data.

In this dataset, instead of training a DNN model first and then applying the interaction scores to the DNN model, we apply the interaction scores to the underlying relationship of the simulation directly. This is equivalent to applying the interaction scores to the most accurate DNN model (although impossible to achieve in reality), which completely recovers the relationship underlying the data from the data. For comparison, we also apply the PDV interaction value method and the GSM interaction coefficient method to the underlying relationship. By applying those methods directly on the underlying relationship, we eliminate the effects due to the DNN model being only an approximation of the underlying true relationship.

Real-world Data. The source of the real-world data is the Veteran Health Administration’s Corporate Data Warehouse, which is accessible through the VA Information and Computing Infrastructure (VINCI). The dataset is derived from a cohort developed by Cheng et al. [24] for the study of Alzheimer’s disease and related dementias (ADRD), and contains 500,000 patients of only two races: black and white. An index date is defined for each patient, which is the date of the first encounter following regular use of the VHA service, defined as at least one hospitalization or two outpatient visits within a period of 12 months. It is also required that the patients were at least 65 years old at the index date and had no diagnosis of ADRD before the index date. Among the 500,000 patients, 88,027 (17.6%) were cases, i.e., the patients who received a diagnosis of ADRD within 10 years following the index date, and 411,973 (82.4%) were controls, i.e., the patients who received no diagnosis of ADRD and did not die within 10 years following the index date. The patients who died within the 10 years are not included in the dataset.

The task for this dataset is to build a DNN model for classifying the patients as cases and controls based on the demographics data and baseline characteristics data. The baseline characteristics data include body mass index (BMI) measured on the index date and a set of chronic comorbid conditions indicated by the ICD-9/10 codes received on or before the index date. Since not all outcomes (e.g., mortality) are considered, we call this task a classification rather than an outcome prediction.

Based on the collected data, we construct 29 variables including two continuous variables and 27 binary variables. (See Table 2 for a full list of these variables.) The two continuous variables are “AGE” and “BMI”, which are standardized to have values between 0 and 1. The 27 binary variables are dummy variables derived from all the remaining characteristic data, which are categorical. For each comorbid condition, the corresponding variable takes value 1 (resp. 0), representing the presence (resp. absence) of the condition. Since the binary variables automatically take value 0 or 1, all variables will have a value between 0 and 1, which will improve the learning of DNN models, compared to using raw values (e.g., age in years).

A DNN model with eight hidden layers is designed. The input layer has 29 nodes corresponding to the 29 variables constructed above. The hidden layers have alternating node counts of 50 and 30. The output layer has only one node, which outputs a single value between 0 and 1 representing the risk of having ADRD within 10 years following the index date. The DNN model is trained on the data, using the same method as the one on the simulation data. Then, population-level impact scores and population-level interaction scores are computed, with value 0 chosen as the reference value for all the variables except "BMI", for which the median is chosen as the reference value.

For comparison, we also calculate the PDV interaction values and the GSM interaction coefficients based on the DNN model.

## 3. Results

The DNN model trained on the Simulation Data #1 achieves a testing AUC of 0.950. We calculated the (population-level) interaction scores, PDV interaction values, and the GSM interaction coefficients based on the DNN model. Their correlations with the ground truth, i.e., the interaction coefficients ($\theta_{m}$), for the 20 variable pairs $\left(x_{j_{m}},x_{k_{m}}\right)$ whose interaction coefficients are nonzero in the underlying relationship, are shown in Table 3.

The perfect sign agreement shows that the interaction scores completely agree with the ground truth in direction, which means they capture the right interaction type. The high Pearson’s correlation and nearly perfect Spearman’s correlation show that the interaction scores have a high agreement with the ground truth in relative values and relative ranks.

Compared to interaction scores, the PDV interaction values have very low correlations with the ground truth. This is because the PDV interactions values are all non-negative by design, so that they represent the strengths of the interactions with the directions ignored (Figure 1). The GSM interaction coefficients have very similar correlations in all three types with the ground truth compared to interaction scores.

In Figure 1, we show the interactions calculated using the three methods as well as the ground truth values on the 20 variable pairs with nonzero interaction coefficients, and also on 20 randomly selected variable pairs with zero interaction coefficients. The total of the 40 terms are labeled by numbers 1~40, with the first 20 pairs being those with nonzero interaction coefficients and reversely ordered by those ground truth values.

We can see that the interaction scores are all approximately zero for the pairs 21~40, which have exactly zero interactions. They are not exactly zero because the DNN model is only an approximation of the underlying relationship. The interaction scores for the pairs 1~5 and 16~20 are larger in magnitude than the interactions scores for the pairs 21~40, which indicates that the DNN model has captured the strong interactions. The interaction scores for the pairs 6~15 are all close to zero, similar to those for the pairs 21~40, which indicates that the DNN model has some difficulty in capturing the medium level interactions.

Compared to interaction scores, the GSM interaction coefficients are almost the same, while the PDV interaction values are more different. However, the PDV method is similar to the interaction scores in giving higher values to the pairs with strong interactions but approximately zero values to the pairs with exactly zero interactions.

In Simulation Data #2, since we have only a total of six possible pair-wise interactions between the variables, we can list the results for all six variable pairs, as in Table 4.

Table 4 shows that both our method and the PDV method correctly identify the three variable pairs with zero interactions and the other three variables with non-zero interactions. Our method further correctly recovers the two interaction coefficients for the two variable pairs $\left(x_{1},x_{2}\right)$ and $\left(x_{3},x_{4}\right)$. The GSM method gives some small non-zero values to the three variables pairs with zero-interactions, but the results are very close to what our method gives. On the other hand, the PDV method does not distinguish positive and negative interactions, and the results are not at the same scale of what the other two methods give even if only the absolute values are considered.

In Table 5, we show the individual-level interaction scores on four randomly selected individuals from Simulation Data #2. The four individuals are given ID# 1~4, and their data for the predictor variables are listed in the columns 2~5. The calculated interaction scores are listed in columns 6~11.

Table 5 shows that the interaction score for $\left(x_{2},x_{3}\right)$ varies over the individuals, which means our method can provide individualized interactions scores. As both the PDV method and GSM method only compute one value for each variable pair, they do not provide individualized interaction values.

In Table 6, we summarize the demographics data and baseline characteristics of the cohort.

The DNN model trained on the real-world data achieves an AUC of 0.740. After choosing a threshold on the predicted risk scores that optimizes classification accuracy, we find that the model achieves an accuracy of 0.728, a sensitivity of 0.605 and a specificity of 0.728.

We first calculate the impact scores for each of the 29 variables constructed in the Methods section. (Table 2).

Next, we calculate the interaction scores. For 29 variables, there are a total of 29 × 28/2 = 406 variable pairs. However, the pairs of the binary variables derived from the same N-category (N >= 3) variables (e.g., marital status, income category) must be excluded from the possible interaction pairs since interactions do not apply to them. The number of such pairs is 1 (ethnicity) + 1 (marital status) + 4x3/2 (income category) + 1 (rural/urban area) = 9. There are two additional pairs which have no interactions because the corresponding features do not co-occur on any individuals. Therefore, the total number of pair-wise interactions is 406 − 9 − 2 = 395. Here, we only list the interaction scores for the variable pairs involving race as examples (Table 7). A positive interaction score indicates that the "joint" impact of the two variables on the ADRD risk is more than the sum of their individual impacts. Conversely, a negative interaction score indicates that the "joint" impact of the two variables on the ADRD risk is less than the sum of their individual impacts.

This result shows that race interacts with age. Specifically, from Table 5 we know that the ADRD risk increases as age increases; however, African Americans have a higher rate of increase in ADRD risk with age than white Americans (see first row of Table 6). This is consistent with the discovery from a prior study by Guland et al. [25].

Another study by Chen et al. [26] discovered that increased BMI was associated with reduced risk of dementia. According to their results, obesity had a slightly stronger protective effect on African Americans than white cases. These discoveries are also in agreement with our results.

For comparison, the PDV interaction values and the GSM interaction coefficients are also calculated. Because this is only for an illustration of their differences, we only list their results on four selected variable pairs (Table 8).

Table 8 shows that all three methods agree on the strength of the interactions: RACE interacts stronger with AGE and with BMI than with DEPRESSION and with CANCER. Our method and the GSM method also agree on the direction of the interaction on the two stronger interactions (with AGE and with BMI). In contrast, the PDV method only gives positive values and the values are not at a similar scale as the other two methods.

## 4. Discussion and Conclusions

Significance: the DNN model’s ability to capture unknown or undefined interactions present in nonlinear relationships is one of its strengths. Such captured interactions may reveal important underlying relationships (e.g., drug–drug, race-risk factor, gene-environment) in biomedical data. A prior explanation of DNN focused on the individual variables’ contributions to outcomes. We developed a novel method called "interaction scores" to measure interactions captured by DNN models, which generalize the interaction coefficients from logistic regression models to the DNN models.

We applied our method to two simulated and one real-world dataset. It is difficult to validate the explanation results on real-world data because the ground truth is generally unknown. In contrast, it is much easier to validate the results on simulated data, because the ground truth can be obtained from the underlying relationship used to generate the data.

In Simulation Data #1, the validation of the results showed a high level of agreement between the interaction scores and the ground truth. (Table 3). A closer look at the interaction scores show that this validation is also influenced by how well the DNN model approximates the underlying relationship. Therefore, in Simulation Data #2, we applied our method directly on the underlying relationship, which was equivalent to assuming that the trained DNN model completely recovers the underlying relationship (which is impossible in reality). The results (Table 4) show that our method recovered the interaction coefficients of all the variable pairs (including those with zero coefficients) except one (i.e., $\left(x_{2},x_{3}\right)$). This is because the interaction for the latter pair is expressed as a product of two nonlinear functions of the two variables (i.e., $5e^{x_{2}}\sin\left(0.5\pi x_{3}\right)$). This type of interaction generates varying interactions on the population, and Table 5 shows that our method is able to calculate individualized interactions.

For the real-world dataset, the interactions between different ADRD risk factors are not well known. However, we still find some published results [25,26] and they are in the same direction as ours. This consistency supports the validity of the interaction scores.

In addition, we note that, given that our method is model-agnostic, it can be applied to any machine learning model with the same input and output format as the DNN models. We focused on DNN because of its increasing popularity and the difficulty clinicians have in understanding the DNN models.

While there are other existing methods for explaining covariate interactions, they all have limitations. To show what the other methods produce and how they compare to our method, we selected the PDV interaction value and the GSM interaction coefficient methods and applied them to all three datasets. The comparison shows that our method has a similar capability in explaining the covariate interactions as the other methods. The limitation mainly comes from how well the DNN model approximates the underlying relationship. If the DNN model does not capture some interaction effects, then no methods can reveal those from the DNN model. Our method also has some advantages over the other methods. The first and most important one is that the interaction score is designed to be a generalization of the coefficient of the interaction term in a logistic regression, which measures the change in log odds. Therefore, our method produces more interpretable results than the other methods, with the PDV method as a typical example, which does not produce values with intrinsic meanings. The GSM method is an exception, in that it produces results with the same interpretation as ours. However, it does not have the second advantage; that is, our method can produce individualized results (through the individual-level interaction scores), while most of the other methods cannot. This will be useful for personalized medicine.

Implication: The ability to quantitatively assess interactions allows for an additional means to explain DNN models. This is particularly important because DNN models are often not linear. Assessing an individual variable’s importance to or impact on the outcome, of course, is very helpful. Interaction score can complement the explanation provided on individual variables.

The assessment of interactions can generate new hypotheses for further investigations. For example, the interaction between race and several comorbidities in the context of incident ADRD is not well documented in the literature and requires further examination. A better understanding of these interactions could lead to improvements in care. It could also support a more personalized medicine approach in ADRD prevention.

Limitations and Future Work: Although simulated data have an advantage over real-world data, the simulated data may not resemble real-world data well. For example, in real-world data, there can be weak to strong correlations among the predictors, but in the simulated data we used such correlations were not considered. In the future, we plan to use simulated data which better resemble the real-world data in terms of the predictor correlations.

There are "uncertainty" issues related to the method, which occur at two levels: 1) data uncertainty, and 2) model uncertainty. We know that the training data are sampled from a much larger data pool, which may even include data from the future that are not yet available. The data uncertainty refers to the randomness coming from the training data as a random sample. This type of uncertainty is typically quantified in the traditional statistical analysis by, for example, confidence intervals and p-values. We may borrow those uncertainty quantifications from traditional statistics to address such uncertainty in our methods. The model uncertainty means that there can be many different DNN models trained on the same dataset, and they will produce different impact/interaction scores. Many factors influence the final trained model, which include the initial randomly assigned weights, the process of training (e.g., the size and order of the mini-batches), and the regularization method (e.g., early stopping, lasso). This phenomenon is not common in traditional statistical analysis, but has received some attention [27,28]. We plan to investigate these uncertainty issues in the context of DNN models in the future.

Our ADRD dataset required patients to either have 10 years of follow-up or to have ADRD diagnosed within 10 years from the date they entered the cohort. This excluded patients who died within 10 years from the index date. The reason for this is because death is a competing risk for ADRD. However, the average life expectancy for certain age groups (e.g., 80–90, and 90+) is less than 10 years. By requiring 10 years of follow-up, we exclude a number of patients in those age groups. We are planning to repeat this analysis in patients with shorter lengths (e.g., 3 or 5 years) of follow-up.

Given that our ADRD project’s goal is not to predict ADRD risk, the dataset included a limited number of variables. We performed minimum tuning of the DNN model for predictive performance. Nevertheless, the AUC of the DNN model is consistent with risk prediction performance reported in the literature [29,30,31]. Note that our follow-up time (10 years) is also longer than most studies in the literature, which may have increased the difficulty in accurately predicting the outcome. In future studies, we may include a larger number of predictors.

We only examined pair-wise interaction. More complex interactions could involve three or more variables. We plan to extend our algorithm in future analysis. There are also many future applications we would like to explore, including identifying novel drug–drug and drug–disease interactions and identifying interaction effects for other clinical domains such as diabetes and cancer. Our goal is to support precision medicine through enhancing clinical data analysis.

## Figures and Tables

**Figure 1 jpm-13-00217-f001:**
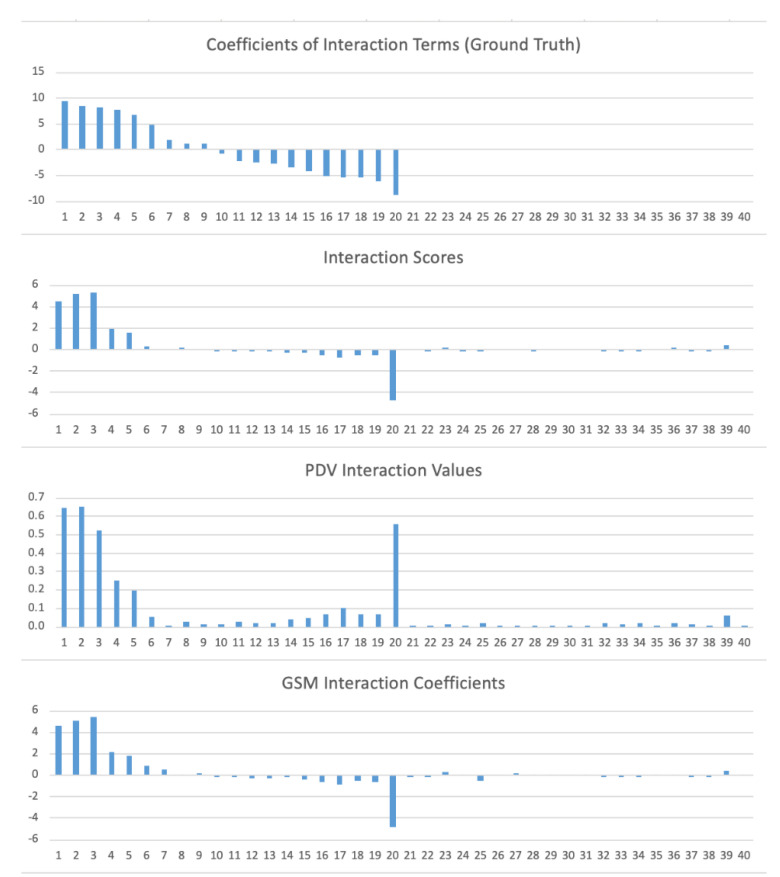
The interactions calculated using the three methods and the ground truth on the 20 variable pairs with nonzero interaction coefficients from the underlying relationship of Simulation Data #1, as well as 20 randomly selected variable pairs with zero coefficients. The horizontal axis is the list of variable pairs, which are labeled by numbers 1 to 40, with the first 20 being those with nonzero coefficients and reversely ordered by their ground truth values.

**Table 1 jpm-13-00217-t001:** Related works on methods for explaining covariate interactions.

Method	Approach	Model-Agnostic	Computational Cost
Intrator’s graphical tool	Visualization	Yes	Low
NIT	Observe Internal Parameter	No	Low
GEH	Train Bayesian neural network	No	Moderate
KL-diff^2^	Local derivatives of the Kullback–Leibler divergence	Yes	Low
SHAP interaction value	Input Perturbation	Yes	High
GSM interaction coefficient	Fit a second interpretable model with linear and interaction terms using the first model’s inputs and outputs	Yes	Moderate
PDV interaction value	Standard deviation of the partial dependence function values	Yes	High

**Table 2 jpm-13-00217-t002:** Full list of variables and their impact scores on the real-world data.

Variable	Impact Score
AGE	2.198
TBI	1.342
DEPRESSION	1.160
STROKE	0.974
BMI	−0.738
CKD	0.693
RACE: Black (vs. White)	0.516
HF	0.483
DIABETES	0.443
ANEMIA	0.407
ETHNICITY: Unknown (vs. Non-Hispanic)	0.372
COPD	0.363
MARITAL_STATUS: Unknown (vs. Married)	0.361
ETHNICITY: Hispanic (vs. Non-Hispanic)	0.285
AFIB	0.248
CANCER	0.151
ASTHMA	−0.130
GENDER: F (vs. M)	−0.105
AREA: Rural (vs. Urban)	−0.097
HLD	−0.071
INCOME_CAT: Q4 (vs. Q1)	−0.066
ARTHRITIS	0.060
INCOME_CAT: Unknown (vs. Q1)	0.052
IHD	0.025
AREA: Unknown (vs. Urban)	0.023
HTN	0.021
INCOME_CAT: Q2 (vs. Q1)	0.017
MARITAL_STATUS: Other (vs. Married)	0.010
INCOME_CAT: Q3 (vs. Q1)	−0.008

**Table 3 jpm-13-00217-t003:** Correlations of the computed (population-level) interaction scores, PDV interaction values and the GSM interaction coefficients with the true interaction coefficients in the underlying relationship for Simulation Data #1. The correlations are based on the 20 variable pairs with nonzero coefficients only.

Correlation Type	Interaction Scores	PDV Interaction Values	GSM Interaction Coefficients
Pearson’s correlation	0.863	0.484	0.889
Spearman’s correlation	0.979	0.186	0.976
Sign agreement	1.0	0.45	1.0

**Table 4 jpm-13-00217-t004:** The interaction scores, PDV interaction values and GSM interaction coefficients applied to the true relationship underlying Simulation Data #2 for all six variable pairs.

Variable Pair	Interaction Score	PDV Interaction Value	GSM Interaction Coefficient
$\left(x_{1},x_{2}\right)$	5.0	1.53	4.90
$\left(x_{1},x_{3}\right)$	0.0	0.0	-0.01
$\left(x_{1},x_{4}\right)$	0.0	0.0	−0.05
$\left(x_{2},x_{3}\right)$	7.24	3.19	6.65
$\left(x_{2},x_{4}\right)$	0.0	0.0	−0.03
$\left(x_{3},x_{4}\right)$	−10.0	4.93	−10.10

**Table 5 jpm-13-00217-t005:** The individual-level interaction scores of four randomly selected individuals from Simulation Data #2.

ID#	Simulated Data				Interaction Scores					
	$x_{1}$	$x_{2}$	$x_{3}$	$x_{4}$	$\left(x_{1},x_{2}\right)$	$\left(x_{1},x_{3}\right)$	$\left(x_{1},x_{4}\right)$	$\left(x_{2},x_{3}\right)$	$\left(x_{2},x_{4}\right)$	$\left(x_{3},x_{4}\right)$
1	−0.71	−0.82	−0.63	−0.31	5.0	0.0	0.0	4.54	0.0	−10.0
2	−0.59	0.76	−0.95	0.34	5.0	0.0	0.0	7.88	0.0	−10.0
3	0.60	0.94	−0.37	0.38	5.0	0.0	0.0	12.28	0.0	−10.0
4	−0.66	0.76	−0.80	−0.16	5.0	0.0	0.0	8.86	0.0	−10.0

**Table 6 jpm-13-00217-t006:** Summary of the real-world data.

	Cases (*n* = 88,027)	Controls (*n* = 411,973)
Age (in years)		
Mean (SD)	74.6 (7.3)	70.1 (5.3)
Race		
Black	11,205 (12.7%)	33,260 (8.1%)
White	76,822 (87.3%)	378,713 (91.9%)
Gender		
Female	1857 (2.1%)	7,317 (1.8%)
Male	86,170 (97.9%)	404,656 (98.2%)
Ethnicity		
Hispanic	3522 (4.0%)	13,227 (3.2%)
Non-Hispanic	81,784 (92.9%)	391,191 (95.0%)
Unknown	2721 (3.1%)	7555 (1.8%)
Marital Status		
Married	52,777 (60.0%)	267,583 (65.0%)
Other (Single/Divorced/Widow)	35,130 (39.9%)	144,178 (35.0%)
Unknown	120 (0.14%)	212 (0.05%)
Income Category		
1st Quartile	22,028 (25.0%)	99,086 (24.1%)
2nd Quartile	21,784 (24.7%)	103,030 (25.0%)
3rd Quartile	21,900 (24.9%)	103,102 (25.0%)
4th Quartile	20,909 (23.8%)	101,253 (24.6%)
Unknown	1,406 (1.6%)	5,466 (1.3%)
Area		
Urban	59,758 (66.7%)	263,318 (63.9%)
Rural	17,070 (19.4%)	92,635 (22.5%)
Unknown	12,199 (13.9%)	55,984 (13.6%)
Body Mass Index (BMI) (in kg/m^2^)		
Mean (SD)	27.9 (5.2)	29.2 (4.9)
Atrial Fibrillation (AFIB)	2514 (2.9%)	8532 (2.1%)
Anemia	2889 (3.3%)	7418 (1.8%)
Arthritis	7807 (8.9%)	40,473 (9.8%)
Asthma	1043 (1.2%)	6447 (1.6%)
Cancer	4738 (5.4%)	21,454 (5.2%)
Chronic Kidney Disease (CKD)	1259 (1.4%)	2,285 (0.6%)
Chronic Obstructive Pulmonary Disease (COPD)	4536 (5.2%)	16,312 (4.0%)
Depression	8071 (9.2%)	18,635 (4.5%)
Diabetes	14,416 (16.4%)	60,163 (14.6%)
Heart Failure (HF)	2080 (2.4%)	4570 (1.1%)
Hyperlipidemia (HLD)	18,559 (21.1%)	117,208 (28.5%)
Hypertension (HTN)	28,081 (31.9%)	155,198 (37.7%)
Ischemia Heart Disease (IHD)	10,652 (12.1%)	51,769 (12.6%)
Stroke	2834 (3.2%)	4983 (1.2%)
Traumatic Brain Injury (TBI)	177 (0.2%)	168 (0.04%)

**Table 7 jpm-13-00217-t007:** List of variable pairs involving race, and their (population-level) interaction scores on the real-world data.

Variable 1	Variable 2	Interaction Score
AGE	RACE: Black (vs. White)	0.6891
BMI	RACE: Black (vs. White)	−0.5328
RACE: Black (vs. White)	ETHNICITY: Hispanic (vs. Non-Hispanic)	−0.2355
ATHRITIS	RACE: Black (vs. White)	−0.1875
AFIB	RACE: Black (vs. White)	0.1851
RACE: Black (vs. White)	AREA: Rural (vs. Urban)	−0.1512
CKD	RACE: Black (vs. White)	0.1474
HTN	RACE: Black (vs. White)	0.1458
STROKE	RACE: Black (vs. White)	0.1366
COPD	RACE: Black (vs. White)	0.1252
ASTHMA	RACE: Black (vs. White)	−0.1190
DIABETES	RACE: Black (vs. White)	−0.0948
RACE: Black (vs. White)	MARITAL_STATUS: Other (vs. Married)	−0.0937
TBI	RACE: Black (vs. White)	0.0898
IHD	RACE: Black (vs. White)	0.0891
RACE: Black (vs. White)	INCOME_CAT: Q2 (vs. Q1)	0.0765
RACE: Black (vs. White)	GENDER: F (vs. M)	0.0713
RACE: Black (vs. White)	INCOME_CAT: Q3 (vs. Q1)	0.0590
ANEMIA	RACE: Black (vs. White)	−0.0501
DEPRESSION	RACE: Black (vs. White)	−0.0493
HF	RACE: Black (vs. White)	0.0259
RACE: Black (vs. White)	INCOME_CAT: Q4 (vs. Q1)	0.0128
HLD	RACE: Black (vs. White)	0.0127
CANCER	RACE: Black (vs. White)	0.0122

**Table 8 jpm-13-00217-t008:** Interaction scores, PDV interaction values and GSM interaction coefficients on four selected variable pairs. The first column is Variable 1, and Variable 2 is always the variable RACE: Black (vs. White).

Variable 1	Interaction Score	PDV Interaction Value	GSM Interaction Coefficient
AGE	0.69	0.057	0.41
BMI	-0.53	0.045	−0.68
DEPRESSION	-0.05	0.005	−0.02
CANCER	0.01	0.006	−0.04

## Data Availability

Population-level aggregated data are available from the authors on reasonable request.

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
