# Peer review of "Enhancing Clinical Data Analysis by Explaining Interaction Effects between Covariates in Deep Neural Network Models"

_jpm, 2023, doi:10.3390/jpm13020217_

Round 1
Reviewer 1 Report
The paper seems to have been passed from a serious rework on it and it handles the very important topic of explaining a neural network decision model according to its inputs. According to the results this is performed successfully and we look forward for the extension of the algorithm to three and more variable interactions.
Some points for consideration:
1. Ref. [1] does not exist in the new, underlined references.
2. ref [8] does not have journal details.
3. what is the relevance of interaction scores to Gini values returned by random forests?
4. what is the relation between interaction vs impact scores?
5. these two sentences seem to contradict each other:
"Impact scores quantify the impact of individual variables on the outcome."
(line 59)
"impact score of a variable does not represent the importance of the
variable and is not intended for variable selection" (line 160)
6. "The hidden layers have alternating node counts of 50 and 30."
How these hyperparameters were decided?
Reviewer 2 Report
In this paper, Shao et al. elaborated a covariate interaction interpretable measurement based on DNN, which is also compatible with other models. In addition, authors confirmed through real-world and simulation data that this algorithm can better interpret clinical data.
Here are some minor suggestions:
1. English expressions need polishing, especially the introduction and discussion sections.
2. Attention should be paid to the uniformity of font and format of the full text.
3. Since this algorithm is not only applicable to DNN, it is necessary to highlight the algorithm itself in the abstract.
